# Insight into the Development of PET Radiopharmaceuticals for Oncology

**DOI:** 10.3390/cancers12051312

**Published:** 2020-05-21

**Authors:** Joseph Lau, Etienne Rousseau, Daniel Kwon, Kuo-Shyan Lin, François Bénard, Xiaoyuan Chen

**Affiliations:** 1Laboratory of Molecular Imaging and Nanomedicine (LOMIN), National Institute of Biomedical Imaging and Bioengineering, National Institutes of Health, Bethesda, MD 20892, USA; lauj2@nih.gov; 2Department of Nuclear Medicine and Radiobiology, University of Sherbrooke, Sherbrooke, QC J1H 5N4, Canada; etienne.rousseau@usherbrooke.ca; 3Department of Molecular Oncology, BC Cancer, Vancouver, BC V5Z 1L3, Canada; dkwon@bccrc.ca (D.K.); klin@bccrc.ca (K.-S.L.); fbenard@bccrc.ca (F.B.)

**Keywords:** positron emission tomography, diagnostic imaging, radiopharmaceuticals, radiochemistry, personalized medicine

## Abstract

While the development of positron emission tomography (PET) radiopharmaceuticals closely follows that of traditional drug development, there are several key considerations in the chemical and radiochemical synthesis, preclinical assessment, and clinical translation of PET radiotracers. As such, we outline the fundamentals of radiotracer design, with respect to the selection of an appropriate pharmacophore. These concepts will be reinforced by exemplary cases of PET radiotracer development, both with respect to their preclinical and clinical evaluation. We also provide a guideline for the proper selection of a radionuclide and the appropriate labeling strategy to access a tracer with optimal imaging qualities. Finally, we summarize the methodology of their evaluation in in vitro and animal models and the road to clinical translation. This review is intended to be a primer for newcomers to the field and give insight into the workflow of developing radiopharmaceuticals.

## 1. Introduction

Positron emission tomography (PET) is a noninvasive nuclear imaging modality that is used, amongst many indications, for diagnosis, staging, and treatment monitoring of cancer [1]. PET imaging is performed by administering, usually intravenously, a positron-tagged radiopharmaceutical into a patient [2]. As the radionuclide decays, emitted positrons travel a short distance before interacting with nearby electrons [2]. The interaction between a positron and an electron leads to an annihilation event, emitting two 511 keV photons in opposite directions [2]. The photons are detected in coincidence, and the signals are reconstructed using computer algorithms to generate 2D projections or 3D images [2]. Widely considered as the most sensitive imaging modality, PET can yield quantitative information such as activity per tissue volume (kBq/mL) versus time or standardized uptake values (SUV), which are more widely used in clinic [3].

The majority of PET scans are performed using [^18^F]fluorodeoxyglucose ([^18^F]FDG), the gold standard of PET radiopharmaceuticals. Leveraging the Warburg effect [4], [^18^F]FDG is taken up by cancer cells with enhanced metabolic and glycolytic rates. However, the uptake of [^18^F]FDG is not cancer specific [5], lowering the specificity of imaging, and also, some cancers have low avidity for glucose as an energy source. Because of this, there is a niche for the development of alternative imaging probes. Furthermore, as radioligand therapy (RLT) continues to cement itself as a prominent treatment strategy [6], PET radiopharmaceuticals can assess drug-target expression across lesions in real time and identify potential responders.

In this review, we discuss the process of developing PET radiopharmaceuticals, focusing on candidate selection, radiochemistry, and preclinical evaluation with the goal of translation. We also briefly summarize essential guidelines by regulatory bodies for preclinical assay of diagnostic radiopharmaceuticals to be performed before clinical trials. We hope this will serve as a resource for the nuclear medicine community and help expedite the progression of radiopharmaceuticals into clinical trials to improve patient care.

## 2. Targeting Vectors

A PET radiopharmaceutical is a pharmacophore that is labeled with a positron-emitting radionuclide. Pharmacophores include, but are not limited to, small molecules, aptamers, amino acids, peptides, antibodies, antibody mimetics, and nanoparticles. These molecules can target cancer biomarkers (e.g., enzymes, receptors, and transporters) or biological processes (e.g., energy metabolism, hypoxia, acidosis, and oxidative stress). The imaging of different cell populations within the tumor microenvironment like cancer-associated fibroblasts and immune cells has also emerged as an area of increased interest. Table 1 provides examples of PET radiopharmaceuticals that have been evaluated in the clinic for various indications. Although these molecules differ in composition, they generally share the following attributes as imaging agents [7]:

**High specificity:** The minimization of off-target binding ensures that sites of uptake are truly representative of the molecular pathology and not of a physiological process. It is not uncommon for small molecule inhibitors to bind promiscuously to other related proteins (e.g., pan-kinase inhibitors [8,9]).

**High binding affinity:** Radiopharmaceuticals should bind with nano- or subnanomolar binding affinity to their target of interest (K_d_ < 50 nM). This increases the sensitivity for receptors/targets that are expressed at low densities and are readily saturable.

**Rapid clearance from nontarget tissues:** PET radiopharmaceuticals clear through the renal pathway, hepatobiliary pathway, or both. The route of elimination depends on multiple factors including size, lipophilicity, charge, and plasma-protein binding. When there is sustained tumor uptake, imaging contrast improves with progressive clearance from blood (Figure 1). In addition, the rapid excretion of radiopharmaceuticals reduces radiation exposure to patients.

**Stability in vivo**: Following administration, radiopharmaceuticals circulate before reaching tumor site(s). The uptake period of a radiopharmaceutical may range from hours (e.g., peptides) to days (e.g., antibodies); therefore, stability against enzymes in plasma or on target tissues is needed. Notably, radiopharmaceuticals that rely on metabolic trapping as a mechanism for retention will not be wholly stable.

**Low immunogenicity or toxicity**: Administered at microdoses, radiopharmaceuticals are unlikely to induce pharmacological or allergic effects. However, the potential for adverse events must still be carefully assessed as some agents will be used multiple times for response monitoring [10].

**Accessible and cost effective:** PET radiopharmaceuticals, including their radiolabeling precursors, should be readily available at low costs and allow for routine clinical use.

Although not a generalizable property, another important consideration in design of PET probes is organ specificity. In many cases, well-differentiated cancers have phenotypes closely related to their organ of origin and designing radiotracers that target normal physiological pathways of those tissues to image in situ or metastatic neoplasia is an attractive strategy. A radiopharmaceutical leveraging this strategy is Na[^124^I]I, which targets well-differentiated thyroid cancer that expresses the Na^+^/I^−^ symporter [11]. Another example is [^18^F]FDGal, targeting the galactose pathway in liver [12]. [^18^F]FDGal can differentiate well-differentiated hepatocellular carcinoma from lesions of other cancers metastatic to liver because [^18^F]FDGal accumulates more readily in hepatocarcinoma. In these cases, organ specificity translates in tumor type specificity, in opposition to general-purpose tracers like [^18^F]FDG that image many types of cancers without allowing easy differentiation between their origin [5].

### 2.1. Probes Based on Bioactive Molecules

Biomimicry is a well-established method for drug discovery and development. Many radiopharmaceuticals are based on nutritional building blocks that sustain life (i.e., carbohydrates, amino acids, fatty acids, and nucleic acids; Figure 2). [^18^F]FDG, regarded as the “molecule of the 20th century”, images the deregulation of cellular energetics—a hallmark of cancers [115]. [^18^F]FDG was conceptualized in the early 1970s by researchers at the National Institutes of Health and the University of Pennsylvania [116,117] and its synthesis was described in 1978 by Ido and colleagues at the Brookhaven National Laboratory [118]. Initially applied for neuroimaging, [^18^F]FDG was quickly adopted in oncology. As a glucose analog, [^18^F]FDG is taken up by cells via glucose transporters and phosphorylated by hexokinase-2 for trapping and retention. The major limitation of [^18^F]FDG is that it is not specific for cancer, as [^18^F]FDG uptake is observed in other inflammatory/infectious diseases [5].

Radiolabeled amino acids (AAs) represent one of the larger classes of PET radiopharmaceuticals based on endogenous molecules. Radiolabeled AAs are typically labeled with ^11^C or ^18^F, and are used in the clinic to image pathologies like brain cancer, prostate cancer, breast cancer, and neuroendocrine tumors [16]. In particular, they are useful for glioma imaging, because they have lower background uptake in the brain compared to [^18^F]FDG [119]. These radiopharmaceuticals are recognized and transported into cells by various AA transporters. Examples of radiolabeled AAs include but are not limited to [^11^C]methionine, L-6-[^18^F]fluoro-3,4-dihydroxyphenylalanine ([^18^F]FDOPA), 2-[^18^F]fluoroethyl-tyrosine ([^18^F]FET), and anti-1-amino-3-[^18^F]fluorocyclobutane-1-carboxylic acid ([^18^F]FACBC). For ^11^C-labeled AAs, the chemical structures are usually unmodified. For ^11^F-labeled AAs, modifications at the AA side chain are common for labeling. Recently, Liu et al. reported a series of ^18^F-labeled AAs by replacing the carboxylate group (–COO^−^) with an isosteric trifluoroborate (BF_3_^−^) group that can be radiolabeled through an isotope exchange reaction [120]. This approach creates a framework for the development of AA radiopharmaceuticals that are analogous to their canonical counterparts (Figure 3). Furthermore, Britton and coworkers developed a method for the electrophilic radiofluorination of unactivated C–H bonds in hydrophobic amino acids (Figure 3) to produce ^18^F-labeled AAs that can visualize glioblastoma and prostate adenocarcinoma xenografts [121,122].

Beyond [^18^F]FDG and radiolabeled AAs, other biomimetic molecules also serve as templates for PET radiopharmaceuticals. [^18^F]Fluorothymidine ([^18^F]FLT), a thymidine analog used to assess tumor proliferation is an example [109]. [^18^F]FLT enters cancer cells via nucleoside transporters for pyrimidine salvage. Once inside the cell, [^18^F]FLT is phosphorylated by thymidine kinase-1 and trapped. [^18^F]FLT has been used to predict response to chemotherapy or radiotherapy in lung, breast, and prostate cancer. [^11^C]Choline is another radiopharmaceutical that is readily taken up by cancers during proliferation. Choline can be converted into phospholipids for cell membrane synthesis. [^11^C]Choline is indicated for patients with suspected prostate cancer recurrence upon elevated blood prostate-specific antigen (PSA) levels following initial therapy and noninformative scintigraphy, computed tomography, or magnetic resonance imaging [90]. Lastly, [^18^F]Fluorodihydrotestosterone ([^18^F]FDHT) and [^18^F]fluoroestradiol ([^18^F]F-FES) represent steroid derivatives that target androgen and estrogen receptors, respectively. They can be used to guide and assess antihormone therapies [24,25,52].

Peptidic or peptidomimetic ligands are among the most diverse groups of radiopharmaceuticals. Many neoplasms overexpress protein receptors that mediate biological processes like proliferation, hormone secretion, angiogenesis, and even metastasis [123,124,125]. The somatostatin receptor type 2 (SSTR2), gastrin-releasing peptide receptor, neurokinin-1 receptor, and cholecystokinin 2 receptor are a few examples of G protein-coupled receptors (GPCRs) that are targets for molecular imaging. The endogenous regulatory peptide is often taken as the lead sequence for development; it is iteratively truncated to identify the minimal sequence required for activity; essential AAs are determined by alanine scanning. Since regulatory peptides are prone to enzymatic degradation, modifications (e.g., D-amino acids, unnatural amino acids, backbone methylation, reduced amine bonds, cyclization) [126], can be introduced to help stabilize the peptide. The radioprosthetic group is generally appended at either the N- or C-terminus of the peptide, with some exceptions (e.g., neuropeptide Y receptor radiopharmaceuticals [127]). The particular success of SSTR2 imaging for neuroendocrine tumors (NETs) serves as the impetus for peptide receptor radionuclide therapy (PRRT) and radiotheranostics (Figure 4).

### 2.2. Probes Based on Drugs

Drugs and drug candidates can be used to develop radiopharmaceutical agents. The primary concern with this approach is how the radiolabel group affects the bioactivity of the drug. For instance, the chemotherapeutic agent 5-fluorouracil was radiolabeled with ^18^F and used to differentiate malignant tissues from inflammatory lesions in preclinical models [130]. The tyrosine kinase inhibitor (TKI) erlotinib was radiolabeled with ^11^C and used in the clinic to assess the mutational status of the epidermal growth factor receptor (EGFR) in a cohort of nonsmall lung carcinoma patients [42]. [^18^F]afatinib, a second-generation TKI targeting EGFR, is undergoing similar clinical investigations [46]. The C-X-C chemokine receptor 4 antagonist plerixafor was directly radiolabeled with ^64^Cu by leveraging its chelating properties as a bicyclam [131]. Except for [^64^Cu]Cu-plerixafor, the chemical structure of the targeting pharmacophore in these examples was left unmodified; an existing atom in their structure was replaced with a radioactive isotope.

In recent years, many major advances in nuclear medicine for cancer imaging come from probes derived from drug candidates. As an example, prostate-specific membrane antigen (PSMA) is a membrane-bound glutamate carboxypeptidase that is expressed in prostate cancer, and neovasculature of solid malignancies [132]. Initial efforts for targeting PSMA led to the development of a radiolabeled monoclonal antibody, [^111^In]In-capromab pendetide [133]. However, [^111^In]In-capromab pendetide was never widely adopted due to its reliance on SPECT technology and the fact that it was directed against an intracellular epitope, resulting in poor sensitivity as only necrotic cells were recognized. In 2001, Kozikowski et al. reported a class of urea-based inhibitors that targeted the enzymatic domain of PSMA [134]. This led to the development of PET-based PSMA radiolabeled with ^18^F or ^68^Ga (Figure 5), with [^68^Ga]Ga-PSMA-11 being the most widely used. These imaging agents are being used to guide RLT in patients with metastatic castration-resistant prostate cancer (mCRPC) in clinical trials [93].

Another biological target with revived interest is the fibroblast activation protein (FAP). FAP is a cell-surface serine protease that is expressed by fibroblast cells. Cancer-associated fibroblasts (CAFs) remodel the extracellular matrix of the tumor microenvironment and facilitate growth and invasion [53]. In 2003, antibody-based imaging of FAP was investigated with [^131^I]I-sibrotuzumab [135], but efforts were hindered by slow pharmacokinetics and poor resolution. A decade later, Jansen et al. identified selective inhibitors of FAP based on a (4-quinolinoyl)-glycyl-2-cyanopyrrolidine scaffold [136]. Based on this work, Lindner et al. were able to develop small molecule FAP-targeting PET imaging agents (Figure 6) [137]. Kratochwil et al. reported that [^68^Ga]Ga-FAPI-04 was taken up by 28 types of cancer, showing itself to be a tumor agnostic agent [54]. The authors are optimizing lead structures to identify companion radiotherapeutic agents [55].

Monoclonal antibodies (mAbs) have been radiolabeled for cancer imaging since the 1970s [138]. However, interest waned briefly due to poor pharmacokinetics, immunogenic responses, low sensitivity and resolution of SPECT, and costs. New developments in oncology have revitalized interest in mAbs or antibody derivatives-based imaging including the clinical success of high-affinity mAbs as cancer therapeutics (e.g., trastuzumab, cetuximab, and bevacizumab) [139], the identification of novel cancer-associated proteins and splice-variants [140], and improved isotope production and radiolabeling strategies [141,142]. Immuno-PET can be a valuable development tool for mAb-based therapies like antibody–drug conjugates. Moreover, immuno-PET can be used in combination with other radiopharmaceuticals to select patients for individualized treatment to improve cost effectiveness [143]. For example, radiolabeled checkpoint inhibitors are being evaluated in the clinic to see if they can be used to predict response for immunotherapy (Figure 7) [100].

### 2.3. Chemical Screens

Chemical screens have been used extensively for drug development, but is not commonly applied in radiopharmaceutical settings, at least not directly. However, it represents a promising strategy that can enable the discovery of new potent ligands with the requisite structure for radioisotope conjugation and the effective pharmacokinetics for imaging. Chemical screens assess the interaction of a library of structurally-diverse molecules to a target of interest; in an ideal scenario, a specific pharmacophore will be identified as having some degree of binding in comparison to others in the library [144]. Chemical screens have evolved from manual bioactivity assays of a group of ligands to techniques that range from high-throughput screens of thousands of compounds [145,146] to computational screens that assess binding affinity between a ligand and a target via docking simulations [147,148].

To generate diverse chemical libraries, a strategy is to use medium- or high-throughput combinatorial chemistry [149]. There is an increasing demand from industry and academia to develop new methods of generating structurally complex compounds in parallel, enabling a greater exploration of the chemical space. Yet, with the advances in combinatorial and synthetic chemistry come logistical challenges. Screening thousands, if not millions, of compounds, inevitably lead to issues as each compound must not only be purified and characterized but also assayed *en masse* [147]. As such, strategies have evolved into assaying mixtures of compounds towards a specific target or system and identifying potent compounds from these complex mixtures.

To identify hits in a complex mixture of compounds, screening strategies include DNA-encoded libraries [150,151,152,153], biological libraries (e.g., phage-display) [154,155,156,157,158,159], and one-bead-one-compound (OBOC) libraries [160,161]. These strategies construct pharmacophores by ligating discrete building blocks of chemicals. The procedure for assaying, separating, and identifying binders differs between methods, and readers are encouraged to further read the directed reviews for a more thorough explanation. Briefly, as pharmacophores increase in length, the diversity of the molecules within the library increases exponentially. For identification, each compound carries a chemical or physical barcode or is identified post hoc via analytical techniques such as mass spectroscopy. For example, binders from DNA-encoded libraries can be identified via their DNA tag through PCR and sequencing [150], and binders from OBOC libraries can be physically displaced from the bead and identified using a chemical tag [162], mass spectroscopy [163], or techniques like Edman degradation for peptides [164].

An example of radiopharmaceutical development from a chemical screen is the development of LLP2A, a pharmacophore with picomolar binding affinity to integrin α_4_β_1_. LLP2A was identified using a diverse OBOC library screen followed by a focused OBOC library screen using Jurkat cells expressing integrin α_4_β_1_ [165]. The diverse library (5.4 × 10^10^ permutations) was designed around the tripeptide LDV, the minimal sequence required for binding to integrin α_4_β_1_ [166] and a 2-(4-(3-*o*-toylureido)phenyl)acetyl N-terminal moiety, previously shown to enhance binding [167]. Screening of this library revealed that (1) unnatural L, D, and V-like amino acids could enhance binding, (2) amino acids proximal of the C-terminus were unnecessary, and (3) 2-methylphenylurea was the ideal N-terminus cap. A focused library with these elements was used in conjunction with a pharmacological blocker and a negative selection control to obtain LLP2A. LLP2A was used successfully for near-infrared fluorescence imaging and later adapted for PET imaging by replacing the fluorophore with a radiometal complex (Figure 8). The resulting radiopharmaceutical, [^64^Cu]Cu-LLP2A, showed good tumor uptake and contrast ratios in preclinical models [168,169] and is currently being investigated in phase I clinical trial for multiple myeloma imaging [74].

Another recent example is the use of a unique dual pharmacophore display DNA-encoded library by Wichert et al. to identify a novel inhibitor of carbonic anhydrase IX (CA-IX) [153]. In brief, 550 unique chemical constructs were conjugated to a 48-mer oligonucleotide, carrying a six-nucleotide sequence for identification, making up Library A. Library B, containing 202 compounds, was then constructed with a nucleotide sequence that could hybridize with Library A and also transfer a unique coding sequence. The oligonucleotide sequence in Library A acted as the template for PCR amplification. CA-IX was fixed on a solid support and the combined library (111,100 compounds) was subjected to affinity capture to identify potential binders. The A-493/B-202 pairing was found to be highly enriched and demonstrated twofold improved binding to CA-IX as compared to acetazolamide, a positive control. Optimization of the spacer between the two pharmacophores yielded a candidate with a K_d_ of 2.6 nM (~10-fold improvement), which was adapted for imaging [153,170,171]. Minn et al. conjugated a bifunctional chelator to the dual pharmacophore for ^64^Cu-labeling and PET imaging [171]. [^64^Cu]Cu-XYIMSR-06 showed excellent uptake in a clear cell renal cell carcinoma xenograft model and high tumor-to-blood, -muscle ratios of >100 within 24 h.

## 3. Radiochemistry

Following conceptualization, identification of a ligand, and design, radiolabeling is the next step in the developmental process for a radiopharmaceutical. Many radioisotopes can be used for PET imaging, as shown in Table 2. Although each of these radioisotopes emits positrons, they have differences in physical half-life, branching ratio, positron range, and chemical reactivity. This raises the question, “How does one select the appropriate radioisotope and radiolabeling strategy for a pharmacophore?” The most straightforward answer would be to select a radiolabel that does not abrogate bioactivity of the pharmacophore. However, there are other considerations such as:

**Positron range**: Positrons emitted from the radioisotope travel a certain distance in the tissue before encountering an electron and causing an annihilation event. The traveled distance is proportional to the positron energy, known as the positron range [172]. This causes uncertainty for determining the exact location from which the positron originated from, reducing spatial resolution. Radioisotopes with lower positron ranges are preferred.

**Suitable physical half-life**: The radioisotope selected for labeling should approximate the biological half-life of the pharmacophore and accommodate clinical logistics if translation is warranted. In particular, ^11^C (t_½_: 20.4 min), ^18^F (t_½_: 109.7 min), and ^68^Ga (t_½_: 67.7 min) are suitable for labeling small molecules, peptides, and affibodies that clear rapidly from circulation on the order of minutes to hours, whereas longer-lived isotopes like ^124^I (t_½_: 100.2 h) and ^89^Zr (t_½_: 78.4 h) are suitable for labeling antibodies, antibody fragments, and nanoparticles that stay in circulation on the order of hours to days. Some isotopes with intermediate half-lives like ^64^Cu (t_½_: 12.7 h) and ^86^Y (t_½_: 14.7 h) are suitable for many types of molecules.

**Molar activity**: Molar activity (MA) is defined as the amount of radioactivity per mole of a compound [173]. For saturable targets, it is better to have radiopharmaceuticals with “high” MA as the unlabeled species can compete for uptake. The concept of “high” MA varies for radionuclides, dependent largely on half-life. For instance, MA >185 GBq/µmol is considered “high” for ^18^F-labeled radiopharmaceuticals [174]. The presence of contaminants (e.g., nonradioactive isotope species) in reaction vessels or tubing reduce MA. Moreover, the production of some radioisotopes may necessitate the addition of a carrier or a stable isotopic species (e.g., [^18^F]F_2_), further decreasing MA [175]. Reactions with low labeling efficiency or that require large amounts of precursor can still achieve high MA, as long as the desired labeled product can be readily separated. Although optimal imaging contrast and tissue uptake is not always correlated with MA [176,177], care should be taken to ensure that injected ligand dose does not exceed practical (clinical) limits [178].

**Ease of labeling**: To minimize synthesis time and radioactive decay, one-step radiolabeling strategies that can proceed with fast kinetics are ideal. Sometimes a reaction can be heated to increase its kinetics, but this strategy should only be applied to pharmacophores that are not heat labile (e.g., small molecules or selected peptides).

**Availability of radioisotope**: Radioisotope selection can be dictated by infrastructure. Radioisotopes can be produced from reactors, accelerators, cyclotrons, and generators [179]. Some radioisotopes can be obtained from multiple production routes [180]. It is possible to ship radiopharmaceuticals (or starting radioactivity) from a central pharmacy (e.g., [^18^F]FDG and [^64^Cu]CuCl_2_), but there are exceptions. If a radiopharmaceutical requires a short-lived isotope (e.g., ^18^C- and ^15^O-labeled compounds, [^82^Rb]RbCl_2_), an onsite cyclotron or generator is required as its half-life prevents distribution.

### 3.1. Radiolabeling Strategies

Strategies for radiolabeling largely fall under two categories: direct labeling and indirect labeling using prosthetic groups. Direct radiolabeling is usually reserved for small molecules as prosthetic conjugation may introduce steric bulk that negates bioactivity. Because carbon and fluorine are ubiquitous elements of organic-based pharmacophores, and ^11^C and ^18^F are readily produced by a cyclotron, they are the two most investigated PET radioisotopes. Precisely, ^11^C is used to label small molecules, particularly those used in brain imaging. Moreover, ^11^C is commonly produced as [^11^C]CO_2_, which can be converted to [^11^C]CH_3_I for methylation or to [^11^C]acyl chloride for amine conjugation [183,184,185]. [^11^C]CH_4_ is another viable synthon for radiolabeling, with equivalent or even higher molar activities [186]. Like [^11^C]CO_2_, [^11^C]CH_4_ can be converted to [^11^C]CH_3_I for methylation [187].

Fluorination strategies rely largely on nucleophilic substitution (S_N_^2^ or SNAr) of [^18^F]fluoride, with an appropriate leaving group (e.g., triflate) appended at the desired site of the molecule (Figure 3). Several metal-catalyzed radiofluorination methods have been developed that can expedite this process [185,188]. Electrophilic-based radiofluorination methods have also been explored. [^18^F]F_2_ has been used for electrophilic radiofluorination; however, given its reactivity and corrosiveness, effort towards milder electrophilic reagents have been pursued [189]. This has led to the development of reagents such as [^18^F]NFSi [190] and [^18^F]F-Selectfluor [191]. These methods are effective for labeling peptides that cannot tolerate nucleophilic substitution conditions [192]. However, these approaches can suffer from low molar activity and thus are better suited for radiopharmaceuticals that target transporters. The chemistry of ^18^F and ^11^C encompasses a field greater than this review can cover and readers are encouraged to peruse the cited reviews for a greater breadth of information.

Pharmacophores that lack a suitable site for direct labeling or cannot withstand the requisite harsh labeling conditions can use a prosthetic group. Prosthetic groups are generally small molecule-based constructs that have been optimized to coordinate radionuclides [193]. For biological considerations, they are appended to a site that is not required for binding (either determined experimentally or predicted using docking/modeling studies). Prosthetic groups are usually attached via a linker which can act as a pharmacokinetic modifier [194]. There are many radioprosthetic groups designed for ^18^F-labeling, and they are selective for different functional groups (e.g., carboxylic acids, thiols, amines, azides, etc.) [195]. The prosthetic group can be labeled first and then conjugated to the pharmacophore (e.g., [^18^F]fluorobenzaldehyde) [196] or it can be first conjugated and subsequently radiolabeled (e.g., silicon-fluoride or trifluoroborate radioprosthetic groups) [197].

Bifunctional chelators are used widely in radiopharmaceutical development to chelate different radiometals [198]. A single chelator may be able to coordinate different radiometals, and a radiometal can be coordinated by different chelators [198]. Given the number of radiometals and chelators that exist, this nearly ensures that there will be a suitable radiometal/chelator combination for any type of pharmacophore [198]. For example, peptides and small molecules can be radiolabeled with ^68^Ga, ^44^Sc, or ^64^Cu using 1,4,7,10-tetraazacyclododecane-1,4,7,10-tetraacetate (DOTA), while mAbs can be radiolabeled with ^64^Cu using 1,4,7-triazacyclononane-1,4,7-triacetic acid (NOTA) or with ^89^Zr using desferrioxamine (DFO) [141]. Like with linkers, chelators can also modulate distribution [199]. Perhaps the most intriguing aspect of bifunctional chelators is that some chelators like DOTA can chelate therapeutic isotopes as well (e.g., ^177^Lu and ^225^Ac). This represents a simple pathway for the development of theranostic pairs for cancer. Moreover, there is the potential to develop dual-labeled radiotheranostics with a chelator for a therapeutic radioisotope and another prosthetic group for a PET radioisotope (e.g., silicon-fluoride or trifluoroborate) [200,201,202].

### 3.2. Automation

Once a lead radiopharmaceutical is identified, the radiosynthesis is automated to optimize time and resource commitments. A synthesis module is usually housed within a hot cell (Figure 9) with operations controlled by an external computer. The module can perform the synthesis, purification, and reformulation automatically, without user intervention. The radiopharmaceutical is automatically dispensed into a presterilized vial. Automation can reduce the potential radiation exposure to personnel, simplify multistep syntheses, and ensure reproducibility and consistency of the radiopharmaceutical (yield, purity, and molar activity) in compliance with current good manufacturing practice (GMP). Moreover, once an automation process is established, the technology can be adopted by other centers with similar equipment, facilitating greater access.

### 3.3. Quality Control

Prepared radiopharmaceuticals must pass quality control (QC) tests before they can be released for patient use. QC for radiopharmaceuticals can be divided into physicochemical or biological tests. Institutes must comply with QC regulations to ensure safety and efficacy. For physicochemical tests, the parameters measured are appearance, radionuclide identity and purity, activity concentration, radiochemical purity, residual solvents, specific activity, pH, osmolality, and stability. For biological tests, the parameters measured are bacterial endotoxin test, filter membrane integrity, and sterility. Readers are encouraged to read the article by Vallabhajosula [203] to see the different methods for testing each parameter. It should be noted that there are regional pharmacopeias that dictate QC methodologies, and that institutions must adhere to a procedure if it is stated in a pharmacopeia. Although most parameters can be tested before release, sterility testing normally has a 2-week delay as it requires incubation in soybean casein digest and fluid thioglycollate medium to test for bacterial and fungal growth; hence, it is not tested before release. Usually, a few production runs are done and once several tests confirm sterility, subsequent runs can be injected in patients as long as bacterial endotoxin and filter membrane integrity pass. Sterility is still assayed knowing that results will only be known long after injection [203,204].

## 4. Preclinical Experiments

The conception of imaging radiopharmaceuticals requires careful planning of preclinical studies to confirm their suitability for the intended application, guide their optimization, and support clinical translation. In this section, we aim to provide a basic list of experiments that should be performed for characterization. Although the order of these steps will vary based on individual laboratory workflows, it is possible to minimize development costs by focusing on critical assays early in the process. In particular, low-cost discriminating in vitro assays like binding affinity and in vitro stability should be done earlier before advancing to in vivo studies.

### 4.1. Binding Affinity

In vitro assays represent an inexpensive and quick method of assaying new probes for bioactivity, while being highly predictive of the potential success of the radiopharmaceutical [205]. A radioligand competition binding assay is generally performed on a cell line or commercial membranes that express the target of interest [206]. The probe will be incubated with a competing ligand. Although either the probe or competitor can be radiolabeled, it is often simpler to use a long-lived commercial radioactive competitor (i.e., ^125^I-labeled). Ideally, the dissociation constant (*K_d_*) of the competitor is known, so that inhibition constant (*K_i_*) can be calculated [206]. Other methods like surface plasmon resonance [207], biolayer interferometry [208], isothermal titration calorimetry [209], and microscale thermophoresis [210] can be used to measure binding affinity. These methods do not require radioactivity, offer higher throughput, and, depending on the method used, inform on binding kinetics; however, they require a lot of optimization for test conditions.

### 4.2. Internalization and Efflux Assays

An important determinant of tumor uptake is cellular internalization and retention of the radiopharmaceutical [205]. Internalization is beneficial since it increases uptake when rates of efflux are low. In particular, radiometals have residualizing properties that are advantageous since once they are internalized, they remain in cell lysosomes (such as ^68^Ga and ^64^Cu) [211,212]. Internalization and efflux assays can be performed to determine those properties. They consist of incubating cells expressing the target of interest with the radiopharmaceutical; thereafter, samples from reaction media or contents of the lysed cells are assayed for their radioactivity at set time points. If desired, a mild acid wash can be applied to cells to determine membrane-bound fraction before lysis.

### 4.3. Stability

Although stability can sometimes be inferred from PET images and biodistribution studies by being attentive to uptake in specific organs (e.g., ^64^Cu in liver, ^124^I in thyroid, ^18^F in bone) [28,104], assays evaluating this specifically are warranted. Instability might be secondary to transchelation, peptide cleavage, dehalogenation, metabolism, and other degradations of the probe. This can be assayed in vitro by incubating the radiotracer in plasma and assaying on radio-HPLC and/or radio-TLC [213]. However, in vivo studies are more accurate since the tracer is exposed not only to plasma enzymes but also to tissue metabolism [214]. At set time points after injection of the tracer in animals, plasma from blood and urine can be collected and then assayed on radio-HPLC and/or radio-TLC to determine the fraction of intact radiotracer. For peptide-based tracers, coadministration with peptidase inhibitors can inform us about enzymatic stability and increasing bioavailability and tumor uptake [215,216].

### 4.4. Plasma Protein Binding

Radiopharmaceuticals may bind to different plasma proteins in blood during systemic circulation. Binding to these proteins can affect a radiopharmaceutical’s half-life and distribution in tissue [217]. If a radiopharmaceutical is lipophilic, it has a higher propensity to associate with plasma proteins [218]. Ultrafiltration is a common experiment used to measure plasma binding [219]. Briefly, a radiopharmaceutical is incubated with plasma or purified proteins (e.g., albumin). The samples are loaded onto ultrafiltration units (commonly 30 kDa cut-off) and centrifuged. The counted eluant represents the fraction of unbound radiopharmaceutical. Though this method is quick and simple, caution should be taken as some radiopharmaceuticals may bind nonspecifically to the filtration membrane. An alternative method to measure plasma binding is high-performance frontal analysis, which is based on gel filtration column chromatography [220]. This technique is said to correlate well with ultrafiltration and minimizes nonspecific binding [220].

### 4.5. Immunoreactivity

Presently, the most common strategy for radiolabeling mAbs for PET is through the use of bifunctional chelators (e.g., desferrioxamine derivatives). These chelators are usually conjugated to the mAb via terminal amines. Since the mAb is modified through the conjugation and radiolabeling process, the immunoreactive fraction may be negatively affected. This reduces binding to the target of interest and in turn, lowers tumor uptake. The Lindmo method has been used extensively to determine the immunoreactive fraction of radiolabeled mAbs [221]. It is a cell-based assay that measures binding at different antigen concentrations and extrapolates to conditions that represent infinite antigen excess. Because it relies on cells, the reliability and robustness of the system are limited by intercellular heterogeneity. Recently, Sharma et al. reported a bead-based radioimmunoassay that can determine the immunoreactive fraction (named “target-binding fraction”), which can be used for radiolabeled antibodies and other targeting vectors as well [222].

### 4.6. Antagonist and Agonist Assays

For some receptor systems like GPCRs, information about whether a radiopharmaceutical behaves as an agonist or antagonist can be important. This property can dictate the extent of internalization of a radiopharmaceutical. A way of determining agonistic/antagonistic property is by analyzing dose–response curves [223]. Briefly, agonists elicit a dose-dependent response, while antagonists do not yield response and can be further assessed by their ability to block the measured signal in the presence of an agonist resulting in a rightward shift of the curve. Partial agonists, while eliciting a response, will only yield a submaximal response as compared to an agonist. Dose–response can also be measured via the release of secondary messengers (e.g., calcium, cAMP, etc.) or by other assays such as FRET/BRET-based assays [224].

### 4.7. Imaging

PET imaging of tumor xenografts implanted in murine models is an attractive method of quickly screening probes [225,226]. Although in vivo studies are inherently more costly, they generate important information indicative of success. Different tumor models include animal, human, or patient-derived xenografts. There is also a trend in nuclear medicine to use transduced models. Transduced tumor models, while convenient for internal comparison and optimization, may not be reflective of the physiological conditions (e.g., PC3 PIP cell line vs. LnCAP cell line used in PSMA research) [178,227]. Other overlooked aspects of in vivo evaluations are the age and gender of the animals. Tumor uptake values in young female mice tend to be higher than in older male mice because of the differences in distribution volume.

Dynamic PET imaging, covering the time from injection of the tracer onward, informs about tracer kinetics (including the time needed to achieve optimal contrast), ability to localize to tumor tissue, clearance routes, and normal tissue uptake. By drawing regions-of-interest (ROIs) on the acquired images, uptake in physiological compartments can be quantified. With short-lived tracers (e.g., ^18^F- or ^68^Ga-labeled), the same animal can be imaged with different tracers on subsequent days, thus optimizing the use of laboratory animals. Animals are typically given *ad libitum* access to food and water; however, there are cases where fasting may be required (e.g., [^18^F]FDG) [228]. Body temperature should be maintained during the uptake period and image acquisition [228]. Pitfalls of using small animals for imaging include partial volume effect [229], which decreases the accuracy of uptake measurements on small organs even when fused with anatomical data (CT or MRI). The use of prolonged anesthesia can also alter pharmacokinetics and clearance of the radiotracer [230]. Nevertheless, PET imaging can be a great screening tool before proceeding with more resource-intensive studies like ex vivo biodistribution.

### 4.8. Biodistribution by Dissection

Biodistribution studies performed by animal dissection are standard practice in the field to determine tracer uptake in tissues. Tumor-bearing animals are injected with the radiopharmaceutical and euthanized at specific time points. Tumor and major organs are harvested and assayed for their radioactivity in an automated gamma counter. Tissue uptake is usually reported as a fraction of injected radioactivity per mass of tissue (%IA/g) decay corrected to the time of injection. Blood is also collected during biodistributions since tracer blood levels can explain observed dynamics. The data can be compared to image-derived biodistribution data, improving scientific rigor.

### 4.9. Specificity

Blocking studies, either in imaging or biodistribution, is an important step to determine radiopharmaceutical specificity for its target. Radiopharmaceutical accumulation in tissue can be driven by different mechanisms that do not involve the target, for instance, enhanced permeability and retention (EPR) [231] effect or passive diffusion. Blocking studies can be performed by coinjecting a 100-fold of the mass of the nonradioactive standard and confirming reduced uptake in the target tissue (at least 30–50% could be considered good). It is also possible to use a different ligand/inhibitor for the target, but its pharmacokinetic properties and affinity must be similar to that of the radiopharmaceutical. Reviewing literature might provide important clues for suitable blocking agents including dose and route of administration. Other ways to test specificity are to evaluate the radiopharmaceutical in models with differing levels of target expression (e.g., a knockout model) and to perform autoradiographic correlation with immunohistochemistry.

### 4.10. Time Points

The selection of appropriate time points is vital for in vivo preclinical studies and is based on expected tracer kinetics, clinical logistics, dosimetry calculations, and radioactive decay. As mentioned, dynamic imaging may inform on appropriate time points for imaging and biodistribution studies. Moreover, time points should be selected that are in line with usual clinical workflows (e.g., 30–120 min post-injection for short-lived isotopes). To determine late pharmacokinetics, late biodistribution time points can be added to coincide with clinical logistics and peak tumor-to-background ratios. Those time points should be spaced over approximately five effective half-lives of the radiotracer. The number of time points recorded should be tailored for the application. If the study is only aimed at comparing uptake between tracers, two or three time points might be appropriate, whereas if it is aimed at determining effective half-life and dosimetry, then enough points should be used for mono- or biexponential decay modeling [232].

### 4.11. Dosimetry

Dosimetry calculations are an essential part of radiopharmaceutical development and are required for clinical translation [232]. It gives an estimation of the radiation dose that a patient receives from the radiopharmaceutical. For most diagnostic probes that incorporate short-lived radioisotopes (e.g., ^68^Ga and ^18^F), it is less a concern since physical decay alone takes care of limiting patient exposure, providing reasonable administered activities. For longer-lived isotopes such as ^89^Zr that are usually associated with antibody or antibody-derivatives imaging, dosimetry can be much higher. To compute dosimetry, according to Medical Internal Radiation Dosimetry (MIRD) methodology, biodistribution studies are typically conducted at multiple time points over five effective half-lives of the radiopharmaceutical [233]. Time–activity curves are fitted to exponential models, and then, the area under the curve is calculated to obtain residence time. These residence times can be entered in specialized dosimetry software like OLINDA/EXM to compute dosimetry. When extrapolating to humans for dosimetry, some adjustments have to be made to account for the organ size difference between species [234]. Because of cost, high-throughput capability, and relative ease of breeding and handling, rodents are the most popular animal model for imaging. Other animal species are also used to evaluate PET radiopharmaceuticals. For example, rhesus monkeys are commonly used to evaluate neuro-PET radiopharmaceuticals with respect to target engagement, distribution, and dosimetry [235].

### 4.12. Toxicity

Evaluation of radiopharmaceutical toxicity is usually performed for the candidate for clinical translation. Typically, toxicity is performed for the nonradioactive standard for diagnostic radiopharmaceuticals. Animals (typically a rodent species) are injected intravenously with a large multiple of the expected human mass of radiopharmaceutical (i.e., 1000-fold) calculated by taking into account specific activity attainable during production and expected radioactivity amount to be administered in humans [236]. A group injected with saline can be used as control. This follows the microdose approach for acute toxicity testing in a single species [236]. Animals are monitored for acute effects in the immediate recovery period and then for a prolonged time for chronic effects (usually 2 weeks). Monitored parameters typically include blood chemistry, complete blood counts, weight, animal behavior, signs of distress, and organ histopathology, but should be adapted for sites of known accumulation of the radiopharmaceutical from biodistribution studies and known toxicities of chemically similar compounds if that information is available.

## 5. Regulatory Considerations

Once a candidate radiopharmaceutical has been optimized and is ready for clinical translation, the next step is obtaining regulatory approval for clinical trials and then, eventually, for marketing authorization, if it is promising. Preclinical evaluations can be costly, time consuming, and resource intensive. Fortunately, regulatory agencies have begun to recognize the unique position of diagnostic radiopharmaceuticals and have drafted guidelines to fast track their evaluation.

### 5.1. Nonclinical Evaluation of Radiopharmaceuticals

An important guideline for nonclinical evaluations is the one published by the International Council on Harmonization (ICH): “ICH guidelines M3(R2) on non-clinical safety studies for the conduct of human clinical trials and marketing authorization for pharmaceuticals.” Generally aimed at pharmaceuticals, this document details toxicity studies and microdose trials (for exploratory clinical trials) [236]. It is widely adopted by international regulatory bodies like the US Food and Drug Administration (FDA), the European Medicines Agency (EMA), and Health Canada.

Under the auspice of ICH, the FDA published a guidance document for industry that is aimed at diagnostic radiopharmaceuticals and which is more relaxed than for typical pharmaceuticals [237]. This is because radiopharmaceuticals for diagnostic studies are generally administered in extremely low doses that are unlikely to have pharmaceutical effects. Furthermore, diagnostic radioisotopes have low emission profiles and are usually short-lived, resulting in relatively low dosimetry. The FDA defines a microdose as a mass of 100 µg or less of the radiopharmaceutical. For proteins, the maximum dose is set at 30 nanomoles or less to account for differences in molecular weight. Recommended studies before initiation of phase I clinical trials include pharmacology extended single-dose toxicity and pharmacokinetics.

Pharmacology includes in vitro and in vivo characterizations. For instance, measurement of affinity to target, biodistribution studies, and dosimetry calculations.

Extended single-dose toxicity can be assayed in a single species (typically in rodents) at a large multiple of expected human doses (i.e., 1000-fold). Animals of both sexes should be included, and the route of exposure should be the same as the one intended for use in patients. In such studies, animals are monitored over 14 days with interim necropsy and parameters monitored include clinical signs, body weight, blood chemistry, hematology, and histopathology. Dose scaling between animals and humans should be adjusted on a milligram per kilogram basis for intravenous and a milligram per square meter basis for oral administration.

Pharmacokinetics (usually performed before phase III) would include absorption, distribution, excretion, and metabolism, in vivo and in vitro. Tests should also be performed to assess the possibility of interactions with other drugs if appropriate, depending on the target.

The EMA has similar guidelines available as a draft [238]: “Guideline on the nonclinical requirements for radiopharmaceuticals, Draft.” It also relaxes requirements for preclinical evaluation of radiopharmaceuticals, in particular for minimal modifications of known radiopharmaceuticals or pharmaceuticals. Those are defined as: radionuclide substitution in a known radiopharmaceutical, addition of a radionuclide to a known pharmaceutical, or minimal change to the nonradioactive part of a radiopharmaceutical.

In the case of a minimal change to the nonradioactive part of a new radiopharmaceutical, the required data before clinical trials include details of pharmacology, pharmacokinetics, and toxicology. For radiodiagnostics that fall under the microdose umbrella, the scope of the required data is less important (≤100 µg, ≤1/100 no-observed-adverse-effect-level (NOAEL), ≤1/100 pharmacological active dose OR ≤500 µg total), maximum five administrations with washout (six or more actual or predicted half-lives between dosing), and each dose (≤100 µg, ≤1/100 NOAEL, ≤1/100 pharmacological active dose) [238].

In all cases, conducted studies should be performed in accordance with good laboratory practice (GLP). In some settings, as close to GLP as possible is accepted. It is important to stress that alternative testing approaches to those detailed in the guidance documents may be acceptable. This should always be discussed with the responsible regulatory bodies to ensure that the planned study design is acceptable.

### 5.2. Exploratory Approaches for First-in-Human Studies

To expedite translation and to reduce time and resources, regulatory bodies are being more receptive to exploratory approaches for first-in-human studies [239]. In 2006, the FDA issued its guidance for exploratory investigational new drug (eIND) studies [240]. Since PET radiopharmaceuticals are administered at subpharmacologic microdoses and present fewer potential risks, they can qualify for eIND studies. By definition, an eIND study describes a trial performed early in phase I, involves very limited human exposure and has no therapeutic or diagnostic intent. eIND studies are conducted before dose-escalation, safety, and tolerance studies. These studies can inform us on a candidate’s mechanism of action (e.g., binding property) and pharmacokinetics. Moreover, eIND can be used to drive selection of a lead product from a group of candidates based on human data rather than relying solely on preclinical data.

### 5.3. Marketing Authorization

The final step in porting a newly developed radiopharmaceutical to clinical use is obtaining marketing authorization from regulatory bodies. This final step will enable the sale and use of the radiotracer in clinical settings outside of clinical trials for the indications for which the marketing authorization was sought. This step is usually performed after extensive phase II/III clinical trials have concluded.

## 6. Perspectives and Summary

The design of radiopharmaceuticals is a complex endeavor (Figure 10). The choice of pharmacophore, radionuclide, radiolabeling strategy, and preclinical experiments, all have an essential role to play in creating a diagnostic radiotracer. Beyond what is discussed, readers should keep in mind other considerations. The identification and validation of a biomarker are just as important as radiopharmaceutical design [241]. As articulated by Nimmagadda et al., this extends to the appropriate selection of tumor models (animal, human, and patient-derived xenografts), implantation strategies (subcutaneous, orthotopic, and spontaneous tumors), and host strains (immunodeficient, immunocompetent, genetically modified, and humanized mice) [241]. These factors influence the outcome of experiments, the ability to compare with literature, and the potential for translation of a clinically useful radiotracer.

Successful discovery and development of a diagnostic PET radiotracer may not be the endgame, however. Many targets are also equally suited for therapeutic applications. With careful design choices, theranostic (i.e., therapeutic and diagnostic) radiopharmaceuticals can be developed. Examples of such success stories include Na[^124^I]/Na[^131^I] for thyroid cancer, [^68^Ga/^177^Lu]Ga/Lu-DOTA-TATE for neuroendocrine tumors, and [^68^Ga/^177^Lu/^225^Ac]Ga/Lu/Ac-PSMA-617 for prostate cancer, with many more in development. With this approach, one can evaluate endoradiotherapy suitability by confirming target expression with the diagnostic radiopharmaceutical of the theranostic pair, enabling personalized medicine.

One fact is certain, molecular imaging is a quickly evolving field. Since 2012, four novel PET radiotracers received FDA approval for cancer indications. This represents one-third of all approved PET radiopharmaceuticals dating back to 1972. We anticipate this accelerating trend to continue as more groups than ever contribute to their development. Recognition by regulatory agencies of the unique status of diagnostic radiopharmaceuticals as well as collaboration between research groups and industry is poised to drive the translation of more of them to clinic. As more radiotracers shine their light on cancer, one thing is certain: the future for PET imaging is bright indeed.

## Figures and Tables

**Figure 1 cancers-12-01312-f001:**
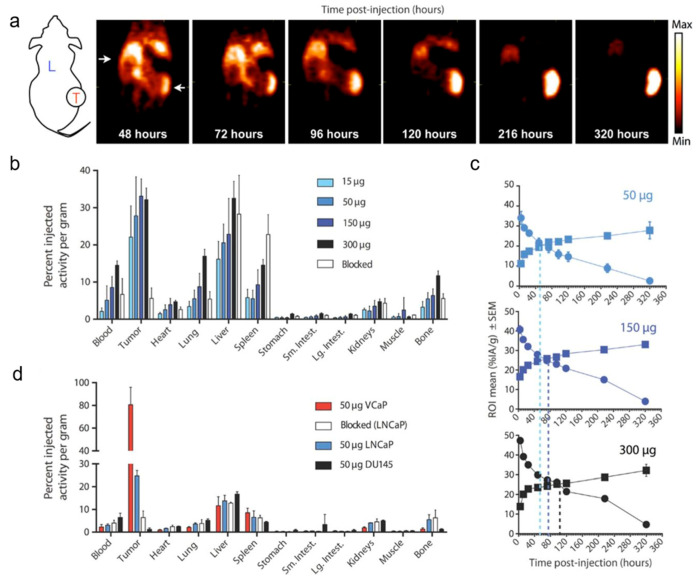
Immuno-positron emission tomography (immuno-PET) imaging of prostate cancer with [89Zr]Zr-11B6. (**a**) Coronal projection images in mice bearing LNCaP tumor xenograft. Longitudinal imaging shows continued uptake in tumor (T) with progressive clearance from liver (L). (**b**) Ex vivo biodistribution of activity in tumor and normal organs at 320 h p.i. (**c**) Time-activity curves in %IA/g of tumors (squares) and blood (circles) for different doses of antibody. (**d**) Greater uptake observed in human kallikrein 2 producing VCaP model compared to LnCaP and nonproducing DU145 xenografts, indicating specificity. Uptake can also be blocked with excess antibody. Figure reproduced with permission from *Sci. Transl. Med.*
**2016**, 8(367): 367ra167 [114].

**Figure 2 cancers-12-01312-f002:**
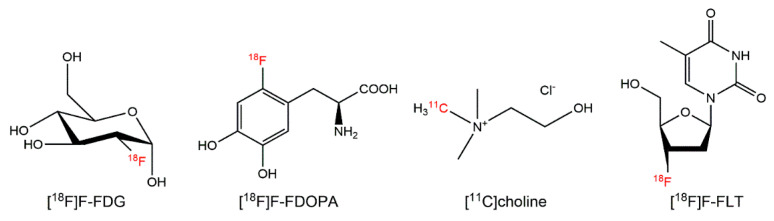
Examples of PET radiopharmaceuticals based on bioactive molecules.

**Figure 3 cancers-12-01312-f003:**
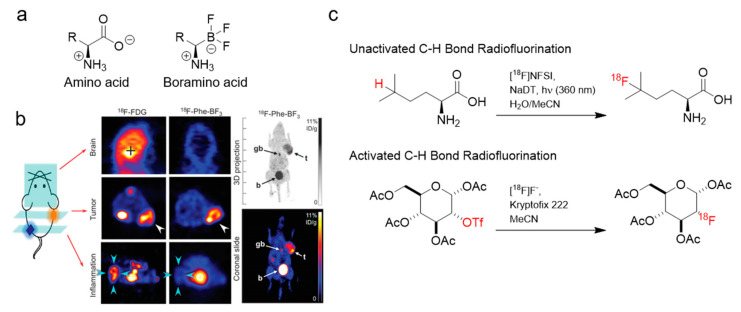
Examples of ^18^F-labeled amino acid derivatives. (**a**) Representative structures of natural amino acids and the synthetic boramino acid variants. (**b**) In vivo PET projection images of [^18^F]FDG and [^18^F]Phe-BF_3_ of the brain, U87MG tumor xenograft, and site of inflammation, respectively. Maximum intensity projection (MIP) images show activity accumulation in tumor, gallbladder, and bladder. Figure adapted with permissions from *Sci. Adv.*
**2015**, 1(8): e1500694 [120], under a Creative Commons Attribution-NonCommerical (CC BY-NC 4.0) License. (**c**) Radiofluorination of an unactivated C–H bond (i.e., lacking a leaving group or an activating proximal functional group) and radiofluorination of an activated C–H bond, via a triflate leaving group for nucleophilic substitution.

**Figure 4 cancers-12-01312-f004:**
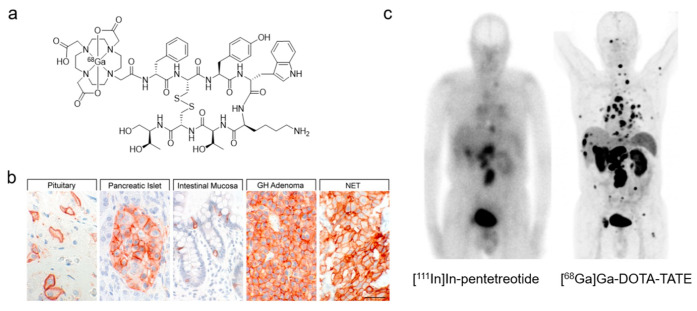
Somatostatin receptor imaging. (**a**) Chemical structure of [^68^Ga]Ga-DOTA-TATE. (**b**) SSTR2 expression in normal tissues and neoplastic tissues. Reproduced with permissions from *Pharmacol. Rev.*
**2018**, 70(4): 763–835 [128]. (**c**) In vivo SSTR2 imaging in a patient with metastatic low-grade cecal NET. [^111^In]In-pentetreotide scintigraphy (left) with [^68^Ga]Ga-DOTA-TATE PET (right) was performed before radiotherapy. In liver, retroperitoneal and thoracic lymph nodes, and bones, PET shows multiple metastases, many of which are undetectable on scintigraphy. Figure reproduced with permission from *J. Nucl. Med.*
**2016**, 57(12): 1949–1956 [129]. Copyright 2016 Society of Nuclear Medicine and Molecular Imaging.

**Figure 5 cancers-12-01312-f005:**
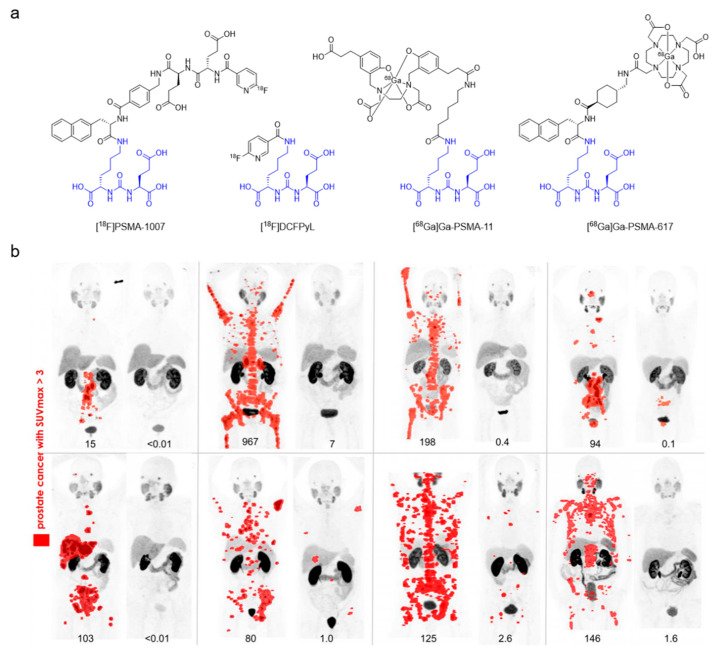
Prostate-specific membrane antigen imaging. (**a**) Chemical structures of several PSMA imaging agents. The four radiopharmaceuticals share a Glu-urea-Lys binding motif (in blue). (**b**) [^68^Ga]Ga-PSMA-11 PET maximum intensity projection (MIP) images at baseline and 3 months after [^177^Lu]Lu-PSMA-617 treatment in eight patients with PSA decline of ≥98% in a prospective phase II study. Lesions with standardized uptake value (SUV) over three are highlighted in red. PSA values (ng/mL) are indicated below MIP images. Figure reproduced with permission from *J. Nucl. Med.*
**2019**, jnumed.119.236414 [93]. Copyright 2019 Society of Nuclear Medicine and Molecular Imaging.

**Figure 6 cancers-12-01312-f006:**
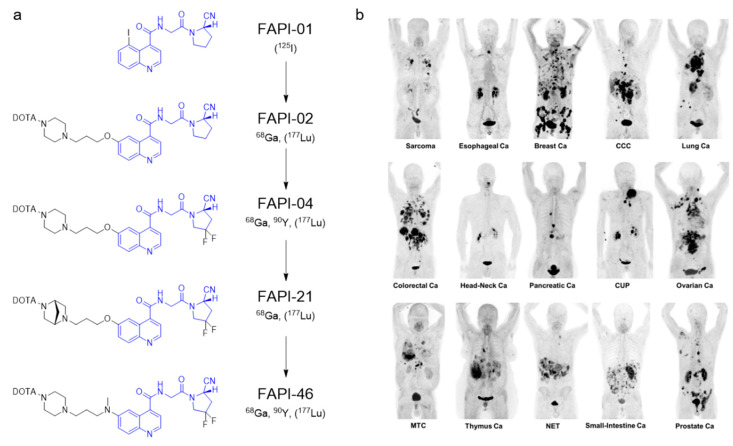
Fibroblast activation protein imaging. (**a**) Chemical structures of FAP-targeted radiopharmaceuticals, which were investigated in detail preclinically and/or clinically. Radionuclides in parentheses were used for preclinical studies. The compounds share a common binding motif (in blue). Figure reproduced with permission from *EJNMMI Radiopharm. Chem.*
**2019***,* 4:16 [53], under a Creative Commons Attribution 4.0 International License. (**b**) Maximum-intensity projection (MIP) images of [^68^Ga]Ga-FAPI-04 PET/CT in patients reflecting 15 different histologically proven tumor entities. Ca = cancer; CCC = cholangiocellular carcinoma; CUP = carcinoma of unknown primary; MTC = medullary thyroid cancer; NET = neuroendocrine tumor. Figure reproduced with permission from *J. Nucl. Med.*
**2019**, 60(6): 801–805 [54]. Copyright 2019 Society of Nuclear Medicine and Molecular Imaging.

**Figure 7 cancers-12-01312-f007:**
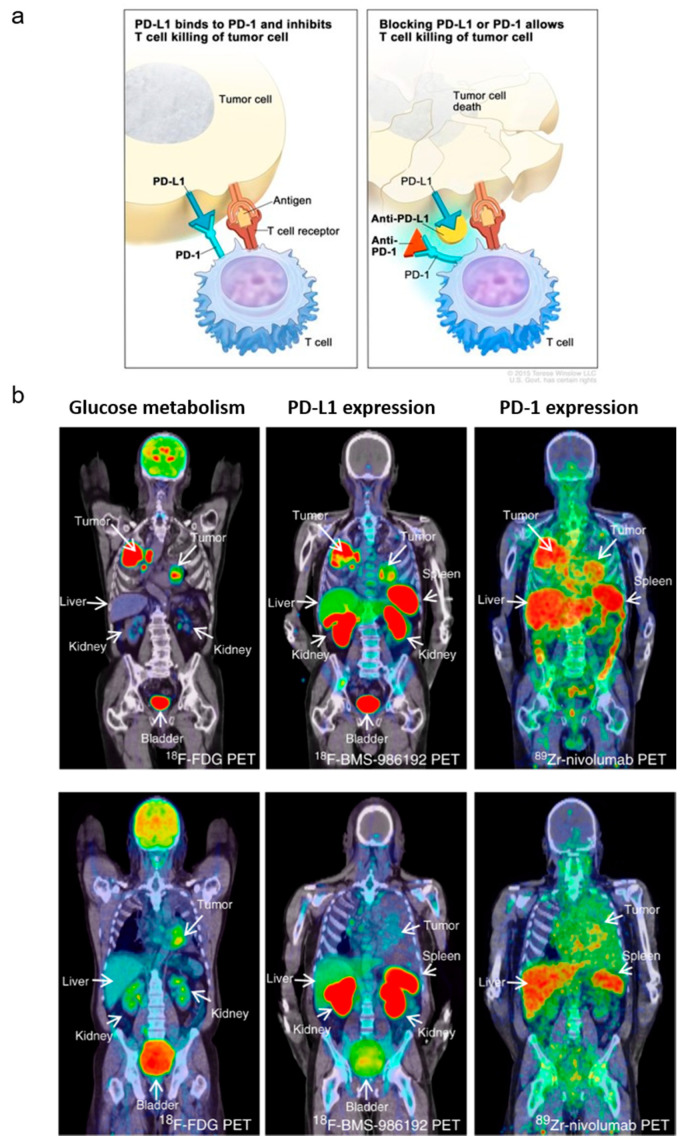
Programmed cell death protein (PD-1)/programmed death-ligand 1 (PD-L1) imaging. (**a**) Blocking the binding of PD-L1 to PD-1 with an immune checkpoint inhibitor (anti-PD-L1 or anti-PD-1) allows T cells to kill tumor cells. Figure courtesy of Terese Winslow for the National Cancer Institute © (2020) Terese Winslow LLC, U.S. Govt. has certain rights. (**b**) PET scans of two patients imaged with [^18^F]FDG that measures glucose metabolism (left), [^18^F]BMS-986192 that measures PD-L1 expression (middle), and [^89^Zr]Zr-nivolumab (right) that measures PD-1 expression. Heterogenous tracer uptake observed between and within lesions. Figure was reproduced with permission from *Nat. Commun.*
**2018**, 9: 4664 [100], under a Creative Commons Attribution 4.0 International License.

**Figure 8 cancers-12-01312-f008:**
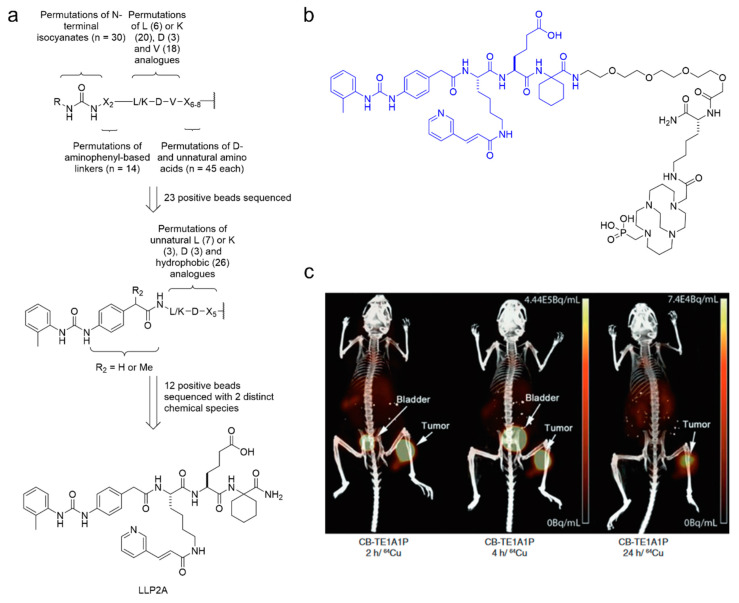
Integrin α_4_β_1_ targeting by LLP2A_._ (**a**) The design of the diverse and focused library targeting integrin α_4_β_1,_ leading to the identification of the LLP2A pharmacophore. (**b**) Chemical structure of LLP2A-CB- LLP2A-CB-TE1A1P, a precursor for ^64^Cu-labeling currently being evaluated in Phase I clinical trials. (**c**) PET/CT images produced by [^64^Cu]Cu-LLP2A-CB-TE1A1P in B16F10 xenograft mice acquired at 2, 4, and 24 h post-injection. Figure adapted with permission from *J. Nucl. Med.*
**2014**, 55(11): 1856–1863 [169]. Copyright 2014 Society of Nuclear Medicine and Molecular Imaging.

**Figure 9 cancers-12-01312-f009:**
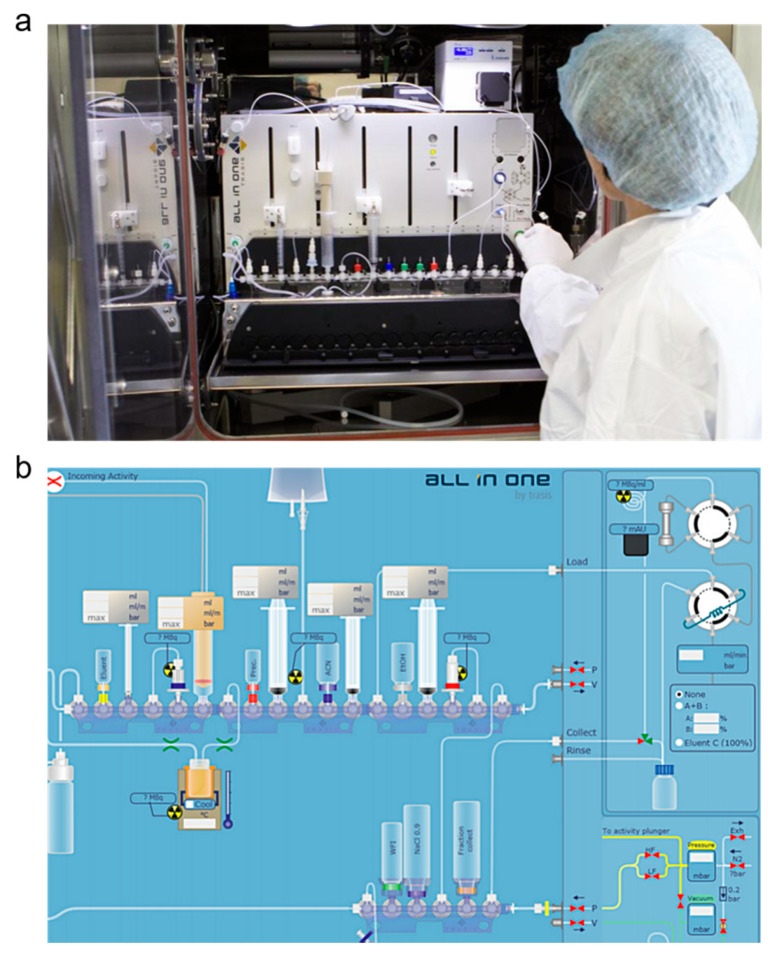
Radiopharmaceutical automation. (**a**) Photograph of a radiochemist setting up an automated synthesis module. (**b**) Example of a graphical user interface for radiopharmaceutical synthesis. Images courtesy of Trasis.

**Figure 10 cancers-12-01312-f010:**
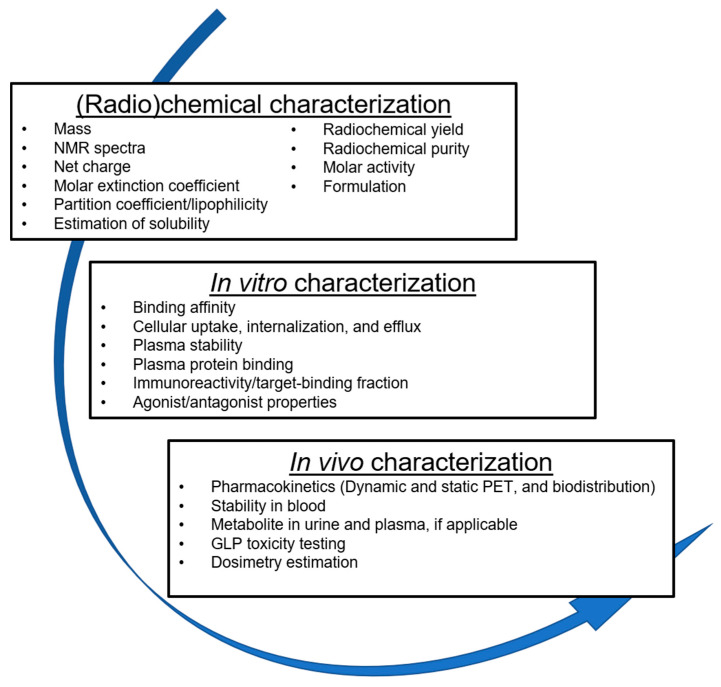
An overview of the radiopharmaceutical developmental pathway. Radiopharmaceuticals undergo comprehensive (radio)chemical, in vitro, and in vivo characterization before they can advance into clinical testing.

**Table 1 cancers-12-01312-t001:** Representative positron emission tomography (PET) radiopharmaceuticals evaluated in clinical studies for oncology.

Biological Process/Target	Radiopharmaceutical	Vector	Indication	References
A33	[^124^I]I-huA33	Antibody	Colorectal cancer	[13]
Acetyl-CoA synthetase	[^11^C]acetate ^#^	Salt	General cancers	[14]
Amino acid transport	[^11^C]methionine ^#^ [^18^F]FDOPA ^#^ [^18^F]FET ^#^ [^18^F]FGln [^18^F]FSPG [^18^F]FACBC * [^18^F]FACPC	Amino acid	Glioma, neuroendocrine tumors, prostate cancer	[15,16,17,18,19,20,21,22,23]
Androgen receptor (AR)	[^18^F]FDHT	Hormone	Prostate cancer	[24,25]
Apoptosis	[^18^F]ML-10 [^18^F]ICMT-11	Small molecule	Glioblastoma multiforme, breast cancer, lung cancer	[26,27]
Bone remodeling	[^18^F]NaF *^,#^	Salt	Osseous lesions	[28]
CA19.9	[^89^Zr]Zr-DFO-HuMab-5B1	Antibody	Pancreatic cancer and bladder cancer	[29]
Carbonic anhydrase 9 (CA-IX)	[^124^I]I-girentuximab [^89^Zr]Zr-girentuximab	Antibody	Clear-cell renal cell carcinoma	[30,31,32]
Carcinoembryonic antigen (CEA)	[^89^Zr]Zr-AMG 211	Bispecific T-cell engager	Gastrointestinal adenocarcinoma	[33]
CD8	[^89^Zr]Zr-Df-IAB22M2C	Minibody	Melanoma, lung cancer, hepatocarcinoma	[34]
CD20	[^89^Zr]Zr-rituximab [^89^Zr]Zr-obinutuzumab	Antibody	B cell lymphoma	[35,36]
CD44v6	[^89^Zr]Zr-U36	Antibody	Head and neck cancer	[37]
C-X-C chemokine receptor type 4 (CXCR4)	[^64^Cu]Cu-plerixafor	Small molecule	Hematological and solid malignancies	[38,39,40]
	[^68^Ga]Ga-pentixafor [^68^Ga]Ga-NOTA-NFB	Peptide		
Cytotoxic T-lymphocyte-associated protein 4 (CTLA-4)	[^89^Zr]Zr-ipilimumab	Antibody	Melanoma	[41]
Epidermal growth factor receptor (EGFR)	[^11^C]erlotinib [^11^C]PD153035 [^18^F]afatinib	Small molecule	Nonsmall cell lung carcinoma; colorectal cancer	[42,43,44,45,46]
	[^89^Zr]Zr-cetuximab [^89^Zr]Zr-panitumumab	Antibody		
Epidermal growth factor receptor 2 (ERBB2)	[^68^Ga]Ga-ABY-025	Affibody	Breast cancer	[47,48,49]
	[^68^Ga]Ga-HER2-Nanobody	Nanobody		
	[^89^Zr]Zr-trastuzumab [^89^Zr]Zr-pertuzumab	Antibody		
Epidermal growth factor receptor 3 (ERBB3)	[^89^Zr]Zr-GSK2849330 [^89^Zr]Zr-lumretuzumab	Antibody	Solid malignancies	[50,51]
Estrogen receptor (ER)	[^18^F]FES[^18^F]4FMFES	Hormone	Breast cancer and gynecologic cancers	[52]
Fibroblast activation protein α	[^68^Ga]Ga-FAPI-04 [^68^Ga]Ga-FAPI-21 [^68^Ga]Ga-FAPI-46	Small molecule	Solid malignancies	[53,54,55]
Galactose metabolism	[^18^F]FDGal	Small molecule	Hepatocarcinoma	[56]
Gastrin-releasing peptide receptor (GRPR)	[^64^Cu]Cu-CB-TE2A-AR06 [^18^F]-BAY 864367 [^68^Ga]Ga-RM2 [^68^Ga]Ga-SB3 [^68^Ga]Ga-RM26 [^68^Ga]Ga-BBN-RGD [^68^Ga]Ga-NOTA-Aca-BBN [^68^Ga]Ga-NeoBOMB1	Peptide	Prostate cancer, breast cancer, glioma	[57,58,59,60,61,62,63,64,65]
Glucagon-like peptide 1 receptor (GLP-1R)	[^68^Ga]Ga-NOTA-exendin-4	Peptide	Insulinoma	[66]
Glucose metabolism	[^18^F]FDG *^,#^	Small molecule	Neoplasm	[1]
Glypican 3	[^124^I]I-codrituzumab	Antibody	Hepatocarinoma	[67]
Hypoxia	[^18^F]EF5 [^18^F]FMISO# [^18^F]FAZA [^18^F]HX4 [^64^Cu]Cu-ATSM	Small molecule	Solid malignancies	[68,69,70,71,72,73]
Integrin α_4_β_1_	[^64^Cu]Cu-LLP2A	Peptidomimetic	Multiple myeloma	[74]
Integrin α_v_β_3_	[^18^F]F-Galacto-RGD [^18^F]F-FPP(RGD)_2_ [^18^F]F-RGD-K5 [^18^F]F-fluciclatide Al[^1 8^F]F-alfatide-I Al[^18^F]F-alfatide-II [^68^Ga]Ga-NOTA-PRGD2	Peptide	Solid malignancies	[75,76,77,78,79,80,81,82]
Integrin α_v_β_6_	[^18^F]F-α_v_β_6_-BP [^68^Ga]Ga-DOTA-SFITGv6	Peptide	Head and neck cancer, lung cancer, colorectal cancer, breast cancer, pancreatic cancer	[83,84,85]
	[^18^F]FP-R_0_1-MG-F2 [^68^Ga]Ga-NODAGA-R_0_1-MG	Cystine knot		
Melanocortin-1 receptor (MC1R)	[^68^Ga]Ga-DOTA-GGNle-CycMSH_hex_	Peptide	Melanoma	[86]
Mesothelin	[^89^Zr]Zr-MMOT0530A	Antibody	Pancreatic ductal adenocarcinoma and ovarian cancer	[87]
Neurokinin 1 receptor (NK1R)	[^68^Ga]Ga-DOTA-SP	Peptide	Glioma	[88]
Neurotensin 1 receptor (NTS1R)	Al[^18^F]F-NOTA-neurotensin	Peptide	Prostate cancer	[89]
Phospholipid synthesis	[^11^C]choline * [^18^F]F-choline ^#^	Salt	Prostate cancer	[90,91]
Poly(ADP-ribose) polymerase 1 (PARP1)	[18F]PARPi	Small molecule	Head and neck cancer	[92]
Prostate-specific membrane antigen (PSMA)	[^18^F]PSMA-1007 [^18^F]DCFPyL [^18^F]DCFBC [^18^F]rhPSMA-7 [^68^Ga]Ga-PSMA-11 [^68^Ga]Ga-PSMA-617 [^68^Ga]Ga-PSMA-I&T	Peptidomimetic	Prostate cancer	[93,94,95,96,97,98]
	[^89^Zr]Zr-HuJ591	Antibody		
Programmed cell death protein (PD-1)	[^89^Zr]Zr-durvalumab [^89^Zr]Zr-nivolumab [^89^Zr]Zr-pembrolizumab	Antibody	Nonsmall cell lung carcinoma	[99,100]
Programmed death-ligand 1 (PD-L1)	[^18^F]BMS-986192	Adnectin	Nonsmall cell lung carcinoma, bladder cancer, breast cancer	[100,101,102]
	[^89^Zr]Zr-atezolizumab	Antibody		
Six-transmembrane epithelial antigen of prostate-1 (STEAP1)	[^89^Zr]Zr-DFO-MSTP2109A	Antibody	Prostate cancer	[103]
Sodium/iodine transporter	Na[^124^I]I	Salt	Thyroid cancer	[104]
Somatostatin receptor 2 (SSTR2)	[^64^Cu]Cu-SARTATE [^68^Ga]Ga-DOTA-TATE * [^68^Ga]Ga-DOTA-TOC *^,#^ [^68^Ga]Ga-DOTA-NOC [^68^Ga]Ga-NODAGA-JR11	Peptide	Neuroendocrine tumors	[105,106,107,108]
Thymidine kinase (DNA replication)	[^18^F]FLT ^#^	Nucleoside	Solid malignancies	[109]
Transforming growth factor-beta (TGF-β)	[^89^Zr]Zr-fresolimumab	Antibody	Glioma	[110]
Vascular endothelial growth factor receptor (VEGFR)	[^89^Zr]Zr-bevacizumab	Antibody	Solid malignancies	[111,112,113]

* Approved by the US Food and Drug Administration (FDA); ^#^ Approved by the European Medicines Agency (EMA).

**Table 2 cancers-12-01312-t002:** Radioisotopes for PET imaging**.** Adapted from Conti and Eriksson [172], Holland et al. [181], and Berger et al. [182].

	Half-Life	Decay Mode	Mean β^+^ Energy [MeV]	Mean Positron Range in Water [mm]	Production Route
^11^C	20.4 min	β^+^ (99.8%)	0.386	1.2	^14^N(*p,α*)^11^C
^13^N	10.0 min	β^+^ (99.8%)	0.492	1.8	^16^O(*p,α*)^13^N
^15^O	2.0 min	β^+^ (99.9%)	0.735	3.0	^15^N(*p,n*)^15^O
^18^F	109.7 min	β^+^ (96.7%)	0.250	0.6	^18^O(*p,n*)^18^F
^44^Sc	4.0 h	β^+^ (94.3%)	0.632	2.4	^44^Ti/^44^Sc generator
^64^Cu	12.7 h	β^+^ (17.6%)	0.278	0.7	^64^Ni(*p,n*)^64^Cu ^67^Zn(*p,α*)^64^Cu
^68^Ga	67.7 min	β^+^ (88.9%)	0.836	3.5	^68^Ge/^68^Ga generator
^82^Rb	1.3 min	β^+^ (81.8%) β^+^ (13.1%)	1.535 1.168	7.1 5.0	^82^Sr/^82^Rb generator
^86^Y	14.7 h	β^+^ (11.9%) β^+^ (5.6%) β^+^ (3.6%)	0.535 0.681 0.883	1.9 2.8 3.7	^86^Sr(*p,n*)^86^Y
^89^Zr	78.4 h	β^+^ (22.7%)	0.396	1.3	^89^Y(*p,n*)^89^Zr
^124^I	100.2 h	β^+^ (11.7%) β^+^ (10.7%) β^+^ (0.3%)	0.687 0.975 0.367	2.8 4.4 1.1	^124^Te(*p,n*)^124^I

β^+^ = positron decay.

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
