# Peer review of "Insight into the Development of PET Radiopharmaceuticals for Oncology"

_cancers, 2020, doi:10.3390/cancers12051312_

Round 1

Reviewer 1 Report

I congratulate Lau et al for such a nice review article. The topic was well conceived and nicely written.

Reviewer 2 Report

The submitted work is a valuable summary of the process of radioactive drug discovery and development written at an introductory level that will be helpful for initiates to the field and experts outside the field of nuclear medicine alike. The writing is generally clear and concise, and I recommend it for publication. The following minor suggestions are intended to clear up only marginal points of potential confusion or improve the flow of the manuscript. 

lines 228-229 The proximal repetition of "structure" reads awkwardly. 

lines 296-301 This section contains several unclear statements and should be rewritten per Coenen et al.'s 2017 NMB article toward standardization of this terminology. The reference to "SA" should be removed, as should the adjective "high" (since this depends to a large extent on the half life of the radionuclide) and the odd choice of a numerical value (37 GBq/µmol), which would be almost embarrassingly low for a tracer labeled with F-18 or C-11. The term "carrier" should be defined, as should "contaminants" if it is to be used precisely. Reactions with large precursor masses can have high molar activities or apparent molar activities if the separation of the desired labeled product is facile. This discussion would also benefit from consideration of numerous literature examples showing that the relationship between optimal uptake in a target tissue, affinity for target and non-target tissues, and molar activity is not always proportional or intuitive. 

line 303 Ambient temperatures are not necessary "minimize synthesis time and radioactive decay," but rather to ensure the biological activity of the resulting compound is not compromised (e.g., with mAbs).  

line 306-307 The sentence structure implies that generator produced isotopes are not produced on a cyclotron. This is of course untrue. The sentence itself establishes murky criteria, since other accelerators and reactors aren't mentioned. 

line 320-321 11C is probably produced directly as CH4 at least as often as CO2. 

Section 4.8 I suggest the authors note the value of comparing ex vivo biodistribution data with in vivo, image-derived biodistribution data as a means of validating and increasing experimental efficiency. 

Section 5.1 A comment only: the formatting of this section made it hard for me to follow and lacked the polish characteristic of other parts of the manuscript. 

Reviewer 3 Report

General assessment of the manuscript:

In the current manuscript, Lau et al. review the pathway for developing PET radiopharmaceuticals for oncology, from targeting vectors, radiolabeling strategies, and quality control, to preclinical experiments and regulatory considerations. The manuscript is well written, covers the most important topics, and provides a good insight into the field of PET radiopharmaceutical development. However, the review does not offer “a guide” as suggested by the title. Rather, it provides “insight” and the title should be adjusted accordingly in order to describe the content of the review precisely.

Overall, I consider the manuscript fit for publication in Cancers after minor revisions:

Introduction

Since the review is intended for newcomers in the field of PET oncology, the authors should provide a brief introduction to how the PET technology works (i.e. emission of positron, annihilation, coincidence detection of gamma rays) as well as the common integration of PET cameras with CT or MR modalities.

The authors should also introduce the reader to the fact that PET is a truly quantitative technique yielding information such as kBq/mL tissue versus time or standardized uptake values (SUVs).

2. Targeting vectors

The authors outline six attributes that the PET radiopharmaceuticals in Table 1 share as imaging agents, attributes that should also be considered in the development of new PET radiopharmaceuticals: High specificity, high binding affinity, rapid clearance from non-target tissues, stability, low immunogenicity or toxicity, and accessible and cost-effective.

The authors should provide examples of PET radiopharmaceuticals for each of these attributes. For example:

Lines 56-57: “It is not uncommon for small molecule inhibitors to bind promiscuously…” – provide examples of small molecule inhibitors.

Lines 63-64: “When there is sustained tumor uptake, imaging contrast improves with progressive clearance from blood” – the authors are suggested to provide images/data that illustrate this.

Lines 67: “The uptake period of a radiopharmaceutical may range from hours to days” – the authors should provide examples of radiopharmaceuticals with different uptake periods.

In connection with “High specificity” and ”Rapid clearance from non-target tissues” the authors should also discuss the benefits of organ specificity for a radiopharmaceutical and provide a suitable example (e.g. [18F]FDG versus [18F]FDGal).

2.1 Probes Based on Bioactive Molecules

Line 100: An illustration that exemplifies the replacement of a carboxylate group (-COO-) with the isosteric trifluoroborate (-BF3-) for isotope exchange reaction would be helpful.

Lines 103-104: Illustrations that exemplify the 18F-fluorination of an unactivated versus an activated C-H bond would be helpful.

2.3. Chemical Screens

In this section the authors provide a large number of references (#135 – 175) relating to the generation of chemical libraries by various strategies. However, only one example is provided in which such a strategy has been used for the development of a radiopharmaceutical for tumor imaging, [64Cu]Cu-LLP2A.

It is strongly suggested that the authors reduce the number of references to the general use of chemical screening and focus more on its use in radiopharmaceutical development and provide additional examples.

Although high-throughput screening of large chemical libraries is a commonly used strategy in drug development, it is not commonly applied in the development of radiopharmaceuticals, at least not directly. The authors should clarify their intent with section 2.3. Is it to show that chemical screening is emerging in the field of radiopharmaceutical development or is it to suggest that such strategy should be considered in the future?

3. Radiochemistry

Line 297: replace “SA” with “MA”.

Lines 306-310: The authors should provide examples of radiopharmaceuticals/radioisotopes that are commonly shipped (e.g. [18F]FDG, [64Cu]CuCl2) and some that can only be used on-site (e.g. 11C and 15O labelled compounds and [82Rb]RbCl).

Lines 324: The authors should provide examples of “an appropriate leaving group” (e.g. –OTf for replacement with [18F]fluoride). It would also be helpful, if the authors provide an illustration of nucleophilic substitution, for example in connection to the illustration of 18F-fluorination of an unactivated C-H bond (see above – section 2.1, line 100-104).

Line 326-327: The authors should also mention [18F]F2 as reagent for electrophilic fluorination as it is more commonly used than [18F]NFSi and [18F]F-Selectfluor.

Lines 377-378: The authors should also mention that there exists several (regional) pharmacopeia that dictates many common methods used in quality control. In fact, it is mandatory to follow a procedure if is stated in a pharmacopeia.

Line 381: replace “in real-time” with “before release”.

4. Preclinical Experiments

Lines 435-436: the authors should provide reference for the statement “This technique is said to correlate…”

Lines 460-479: the authors should also discuss choice of animal species in relation to translation of data to humans and also in relation to estimation of dosimetry.

Lines 481-487: the authors should mention that biodistribution can also be assessed by in vivo successive whole-body PET imaging.

 Line 519: replace “MIRD” with “Medical Internal Radiation Dose (MIRD)”.

5. Regulatory Considarations

Line 589: write out "GLP".

Table 1

It is difficult to see which radiopharmaceutical, vector, indication, and reference belong to which biological processes/targets. There should be a more clear separation between the various biological processes/targets, e.g. by horizontal dashed lines, in the table.

Figure 3 legend

Legend, line 194: “SUV” should be replaced by “standardized uptake value (SUV)”

Figure 4 legend

Line 200: “brackets” should be replaced by “parentheses”.
